PTX3 mediates PI3K/AKT/mTOR signaling to downregulate apoptosis and autophagy to attenuate myocardial injury in sepsis

Cui Na 1
Chen Zhi 2
Yu Zhanbiao 1
Lv Xiaowei 1
Hu Zhenjie 3 hbbdcn2023@163.com
1 Department of Critical Care Medicine, Affiliated Hospital of Hebei University , Baoding , China
2 Hepatobiliary Surgery Department, Affiliated Hospital of Hebei University , Baoding , China
3 Department of Critical Care Medicine, The Fourth Affiliated Hospital of Hebei Medical University , Shijiazhuang , China
Wang Jincheng
Electronic publication date: 2024 May 20
Publication date: 2024
Volume: 12
Electronic Location ID: e17263
Received 2023 Oct 13; Accepted 2024 Mar 28
Copyright: © 2024 Cui et al.
Copyright year: 2024
Copyright holder: Cui et al.
License: This is an open access article distributed under the terms of the Creative Commons Attribution License, which permits unrestricted use, distribution, reproduction and adaptation in any medium and for any purpose provided that it is properly attributed. For attribution, the original author(s), title, publication source (PeerJ) and either DOI or URL of the article must be cited.
License URL: https://creativecommons.org/licenses/by/4.0/

Keywords: PTX3, Myocardial injury, Apoptosis, Autophagy

Funding: The authors received no funding for this work.

==============================
Background

This study aimed to investigate the effect and mechanism of Pentraxin 3 (PTX3) on myocardial injury in sepsis.

Methods

Thirty male C57BL/6 mice were randomly assigned to Groups A, B, or C. Mice in Groups A and B were injected with unloaded lentivirus, while mice in Group C were injected with lentivirus encoding PTX3 overexpression. Seven days after injection, septic myocardial injury mouse models were constructed following intraperitoneal injection with LPS in Groups B and C, and mice in Group A were intraperitoneally injected with normal saline. Cardiac function was examined using echocardiography; pathological variation of myocardial cells was measured through HE staining, transmission electron microscopy, and TUNEL staining; and Western blot was used to measure the expression of PI3K/AKT/mTOR pathway-related, autophagy-related, and apoptosis-related proteins in mice myocardial cells.

Results

PTX3 significantly improved cardiac function and structure in sepsis-stricken mice, and PTX3 alleviated cardiac damage caused by sepsis. PTX3 reduced the relative protein expression of p-PI3K, p-AKT, mTOR, LC3I/II, Beclin, ATG5, Bax, Caspase-3, and Caspase-9 in septic mouse cardiomyocytes and increased the relative protein expression of Bcl-2.

Conclusion

PTX3 can attenuate myocardial injury in sepsis due to the down-regulation of apoptosis and autophagy induced by the PI3K/AKT/mTOR pathway.

Introduction

Sepsis is defined as an organ dysfunction caused by a host’s dysfunctional response to infection, and its prevalence and mortality in clinical management continues to escalate (Huang, Cai & Su, 2019). As a primary organ vulnerable to sepsis-related damage, the heart frequently succumbs to myocardial injury due to sepsis (Bi et al., 2022). Myocardial injury in septic patients is predominantly characterized by systolic and diastolic dysfunction of the left ventricle (Chu et al., 2019). Myocardial damage afflicts more than 40% of individuals with sepsis, and contributes to over 70% of mortalities associated with the condition (Ge, Liu & Dong, 2018).

Apoptosis and autophagy function as pivotal catalysts in the onset of sepsis (Bi et al., 2023). In cardiomyocytes, sepsis can induce autophagy, a condition that undergoes dynamic shifts that are parallel to the progression of sepsis (Sun, Cai & Zang, 2019). However, mitigating cardiomyocyte autophagy can attenuate sepsis-induced myocardial damage (Yang et al., 2021). In a similar vein, cardiomyocyte apoptosis plays a significant role in the pathological cascade that culminates in myocardial injury during sepsis (Fu et al., 2022). Nonetheless, evidence suggests that curbing cardiomyocyte apoptosis may offer therapeutic benefits in mitigating myocardial damage within the septic milieu (Li et al., 2019). Both cellular autophagy and apoptosis are modulated by the PI3K/AKT/mTOR signaling cascade (Peng et al., 2022), providing therapeutic leverage through the regulation of its constituent proteins to ameliorate the pathological state of myocardial injury in sepsis (Xie et al., 2020).

Pentraxin 3 (PTX3), a constituent of the pentraxin superfamily, has been identified as a homologous molecule of C-reactive protein (Ristagno et al., 2019), which is heralded as a putative biomarker of inflammation and an innate immunity modulator associated with immune evasion (Zhang et al., 2022). PTX3 functions as a key regulator in the inflammatory nexus of sepsis (Tian et al., 2019). Consequently, this investigation delves into the apoptosis and autophagy engendered by the PI3K/AKT/mTOR axis through both in vivo and in vitro studies, and provides an exhaustive portrait of PTX3’s influence on myocardial injury in sepsis and its underpinning mechanisms.

Materials and Methods

Lentiviral vector preparation

The oligonucleotide fragments containing PTX3 primer sequences (PTX3-F:5′-AGGTCGACTCTAGGATCCCGCCACCATGCATCTCCTTGCGATTCTG-3′,PTX3-R:5′-TCCTTGTAGTCCATGGATCCTGAAACATACTGAGCTCCTCC-3′) were cloned into lentiviral expression vectors (A35684, Thermo Fisher Scientific CN, Shanghai, China). After extraction, they were transfected with both overexpressing PTX3 lentiviral vectors and control lentiviral vectors, and then the medium and culture were changed 24 h after transfection.

Animal grouping and modeling

Thirty SPF male C57BL/6 male mice (4–6 weeks, 18 ± 1 g) provided by the Beijing Vital River Laboratory Animal Technology Co., Ltd. were adaptively fed for 1 week, during which time all mice had available foodstuff and water. All procedures involving animals conformed to National Research Council (US) (2011) and were approved by the Experimental Animal Ethics Committee of Hebei University. Mice were assigned to groups according to their weight and cage order. Mice were housed in SPF-level animal houses, with 12 h alternating light-dark cycles, temperature maintained at 25 °C, relative humidity maintained at 50%, food and water freely ingested, and experiments started 1 week after the mice acclimatized to the environment. After 1 week, mice were divided into three groups (A, B, and C) with 10 mice in each group via random number table. Mice in Groups A and B were injected with a single dose of unloaded vector (1 × 107 PFU/each), while mice in Group C received injections of a single dose of lentiviral vector encoding overexpression of PTX3 (1 × 107 PFU/each). After 1 week, septic myocardial injury mouse models were constructed from mice in Groups B and C following intraperitoneal injection of LPS (20 mg/kg, SMB00704, Merck, Rahway, NJ, USA). After intraperitoneal injection of LPS, the mice showed signs of sepsis such as malaise, hair erection, and diarrhea. Once the model was established, postoperative care and monitoring were required to ensure the health and welfare of the animals. This included providing plenty of water and food, ensuring that wounds healed well, and minimizing any pain or discomfort. The mice of Group A were intraperitoneally injected with an equal volume of saline. Groups B and C were compared with Group A. After the experiment, mice were euthanized with 20% isoflurane, which induced unconsciousness in the laboratory animals in the shortest possible time in order to reduce pain and fear as much as possible. The mice were judged dead half an hour after their heartbeats stopped. The remaining mice at the end of the experiment were treated with the euthanasia steps described above.

Cardiac function test

All mice were anesthetized with isoflurane (5%, 0.8 L/min), placed on a constant temperature test plate 12 h after modeling, and kept until they were breathing steadily, their voluntary activities were reduced, their muscle tone weakened, and muscles were relaxed. Cardiac ultrasounds were examined with an MX250 ultrasound probe under the Vevo2100 high-resolution small animal ultrasound imaging system, and the probe was placed in the short axis papillary muscle section of the left heart of the mouse to obtain M-mode echocardiography. The probe should not overcompress the chest wall of the mouse so as not to cause respiratory and cardiac arrest, and the ratio of the left ventricular ejection fraction (LVEF), left ventricular shortening fraction (LVFS), left ventricular anterior wall (LVAW), left ventricular internal diameter in diastole (d) (LVID), and E/A ratio were used to evaluate the changes in cardiac function in mice.

Cardiac pathological changes test

HE staining

First, the mice were anesthetized with isoflurane (5%, 0.8 L/min) via intraperitoneal injection, and the apices were excised after removing the mice hearts. The sliced heart tissue was placed in an embedding box and then 4% paraformaldehyde fixative solution, then dehydrated, made transparent, and sectioned. The thickness of each slice was 3–4 um. We placed the slide rack containing tissue slides in a 60 °C oven for 45 min, deparaffinized two times for 15 min each time, passed through absolute ethanol for 10 s, 95% ethanol for 10 s, 80% ethanol for 10 s, and then the slides were washed thoroughly once with clean water. The samples were stained by hematoxylin staining solution (DH0045, Beijing Leagene Biotechnology Co., Ltd., Beijing, China) for 2 min, and then washed with running water for 3 min. Hydrochloric acid ethanol solution was utilized to differentiate and the samples were washed again. Eosin staining solution (DH0045, Beijing Leagene Biotechnology Co., Ltd., Beijing, China) was used to stain for 1 min, then samples were washed with running water, routinely dehydrated, cleared via xylene, and mounted using neutral gum. Finally, the morphological structure of myocardium was observed and photographed under a microscope.

Transmission electron microscope

Myocardial tissues sized approximately 1 mm3 from mice hearts were washed with PBS. After fixation with 4% glutaraldehyde (R0651, Shanghai Yuanmu Biotechnology Co., Ltd., Shanghai, China) and 1% osmic acid (GP18456, Beijing Technology Co., Ltd., Beijing, China), samples were dehydrated with gradient ethanol: 30% ethanol for 15 min, 50% ethanol for 15 min, 70 ethanol uranyl acetate overnight, 80% ethanol for 15 min, 90% ethanol: 90% acetone (1:1) for 15 min (all steps were performed at 4 °C), then treated with 100% acetone for 15 min × 3 h (room temperature), pure acetone and embedding solution (1:1) for 2 h (room temperature), pure embedding solution in a 37 °C oven for 3 h, stained with uranyl acetate (GZ02625, Beijing Technology Co., Ltd., Beijing, China) and lead citrate (GA10701, Beijing Technology Co., Ltd., Beijing, China). The ultrastructural variations of cardiac myocytes were examined by transmission electron microscope (H-7500, Hitachi Limited, Tokyo, Japan).

TUNEL staining

The heart tissue was deparaffinized with xylene, dehydrated with gradient ethanol, and 0.1% Triton X-100 was prepared with 0.1% sodium citrate, 50 μl was added dropwise to each section, which were then rinsed with PBS three times for 5 min each time after 8 min at room temperature. Enzyme solution and label solution were prepared 1:9 on ice to make TUNEL reaction. We added DAPI dropwise until the tissue was completely covered, and counterstained in the dark for 3 min. The specimens were rinsed three times with PBS for 5 min each. The slices were removed and fluorescent quencher was added dropwise to seal the slices. The staining effect was observed under a microscope and photos were taken.

Examination of protein expression level in myocardial tissue

Protein expression in myocardium was examined via Western blot. The cells were lysed with RIPA lysate (P0013K, Jiangsu Beyotime Biotechnology Co., Ltd., Jiangsu, China). SDS-PAGE was utilized to electrophorese the proteins, which were then transferred to PVDF membrane after finishing. The PVDF membrane was then immersed in TBST solution containing 5% skim milk for 1 h, and all antibodies, including PTX3 antibody (ab90806, Abcam, Cambridge, UK), p-PI3K antibody (ab182651, Abcam, Cambridge, UK), t-PI3K antibody (ab302958, Abcam, Cambridge, UK), p-AKT antibody (ab38449, Abcam, Cambridge, UK), p-mTOR antibody (ab109268, Abcam, Cambridge, UK), t-mTOR antibody (ab134903, Abcam, Cambridge, UK), LC3I antibody (ab232940, Abcam, Cambridge, UK), LC3II antibody (ab232940, Abcam, Cambridge, UK), Beclin antibody (ab207612, Abcam, Cambridge, UK), ATG5 antibody (ab109490, Abcam, Cambridge, UK), Bcl-2 antibody (ab182858, Abcam, Cambridge, UK), Bax antibody (ab32503, Abcam, Cambridge, UK), Caspase-3 antibody (ab32351, Abcam, Cambridge, UK), Caspase-9 antibody (ab202068, Abcam, Cambridge, UK), and DAPDH antibody (ab8245, Abcam, Cambridge, UK) were diluted with TBST solution at a dilution ratio of 1:1,000, then were placed into the corresponding PVDF membrane and incubated at 4 °C. The membrane was washed with TBST solution three times continuously the following day, then added into the corresponding secondary antibody chamber for 1-h incubation, and washed again with TBST solution three times. Color was developed with ECL and photographs were taken.

Statistic analysis

All experiments were re-performed three times, and the data were collated using Excel software and analyzed by SPSS 23.0 software. Measurement data were shown as (X ± S), two samples were compared through mean t-test, and analysis of variance was used for comparison between multiple groups. The data satisfied the assumptions of the statistical method, and if the data did not meet the requirements, it was directly deleted. P < 0.05 was considered statistically significant.

Results

PTX3 can ameliorate cardiac injury caused by sepsis in mice

Echocardiography showed significantly lower EF, EF, LVAW, LVAW, d, E/A ratio and significantly higher LVID in groups B and C relative to group A (t-test showed P < for 0.05). Relative to group B, EF, FS, LVAW, d, and E/A ratios were significantly higher and LVID was significantly lower in group C (t-test showed P < for 0.05). PTX3 was shown to improve cardiac impairment caused by sepsis in mice (Fig. 1).

Figure 1 PTX3 ameliorates sepsis-induced cardiac impairment in mice.

(A) Echocardiography of mice in each group; (B) Statistical results of LVEF, LVFS, LVAW, LVID and E/A ratios of mice in each group. Data are presented as mean ± SEM. N = 10; *P < 0.05; **P < 0.01.

PTX3 improved the variation of mice myocardial injury induced by sepsis

HE staining suggested that the mice myocardial cells in Group A were arranged regularly without pathological damages such as necrosis, congestion, or edema. In Group B, the myocardial cells were disorganized with ruptured sarcolemma and larger nuclei observed and simultaneously accompanied by significant necrosis, interstitial congestion, edema, and infiltration of inflammatory cells. Mice in Group C had a significant improvement of the above pathological changes with neatly arranged myocardial structure and a small amount of local necrosis, congestion, and edema. The results of transmission electron microscopy demonstrated that the myocardial fibers in Group A were arranged in order without inflammatory cell infiltration outside the blood vessels. Group B showed myocardial fiber breakage, sarcomere disruption, condensation, and dissolvement as well as blur in some myofilaments and infiltrated mononuclear cells in the myocardial interstitium. In Group C, the broken myocardial fibers were partially repaired, and the sarcomere was arranged neatly. TUNEL staining showed that the number of myocardial apoptotic cells in Groups B and C was significantly increased compared with Group A (t-test showed P < 0.05). Compared with Group B, the number of myocardial apoptotic cells in Group C was significantly reduced (t-test showed P < 0.05). These results indicated that PTX3 improved the variation of mice myocardial injury induced by sepsis (Fig. 2).

Figure 2 PTX3 improved the variation of mice myocardial injury induced by sepsis.

(A) Myocardial HE staining results of mice in each group; (B) Transmission electron microscope observation of mice in each group; (C) TUNEL staining results and apoptotic cell number statistics. Data are presented as mean ± SEM. N = 10; **P < 0.01.

PTX3 inhibited the activation of PI3K/AKT/mTOR pathway in mice myocardial injury induced by sepsis

Western blot analysis showed that the relative protein expressions of PTX3, p-PI3K, p-AKT, and p-mTOR in Groups B and C were significantly higher than those in Group A (t-test showed P < 0.05). Compared with Group B, the relative protein expression of PTX3 in Group C was significantly increased, and the relative protein expression of p-PI3K, p-AKT, and p-mTOR was significantly decreased in Group C (t-test showed P < 0.05). These results demonstrated that PTX3 inhibited the activation of the PI3K/AKT/mTOR pathway in mice myocardial injury induced by sepsis (Fig. 3).

Figure 3 PTX3 inhibited the activation of PI3K/AKT/mTOR pathway in mice myocardial injury induced by sepsis.

(A) Protein band plots of PTX3, p-PI3K, t-PI3K, p-AKT, t-AKT, p-mTOR and t-mTOR; (B) Relative protein expressions of PTX3, p-PI3K, p-AKT and p-mTOR. Data are presented as mean ± SEM. N = 10; **P < 0.01.

PTX3 decreased autophagic capacity of cardiomyocytes in septic mice myocardial injury

Western blot results showed that the relative protein expressions of LC3I/II, Beclin, and ATG5 in Group B were significantly higher than those in Group A and C (t-test showed P < 0.05), while the relative protein expressions of LC3I/II, Beclin, and ATG5 in Group C were significantly higher than those in Group A (t-test showed P < 0.05). Transmission electron microscopy showed that Group A had plenty of well-arranged mitochondria, smooth and integrated mitochondrial cristae, mitochondrial double membrane without damage nor dissolution, vacuoles, and autophagy bodies. In Group B, the number of mitochondria increased and were arranged in disorder, and a number of mitochondria presented with swelling, rupture, and vacuolation, as well as rupture and dissolution in the cristae and increased autophagy bodies. Mitochondria in Group C were lower in number with a relatively neat arrangement and reduced swelling, dissolution, and vacuoles. Furthermore, its cristae was intact and the number of autophagy bodies was fewer. This indicated that PTX3 decreased the autophagic ability of cardiomyocytes in septic mice myocardial injury (Fig. 4).

Figure 4 PTX3 decreased autophagic capacity of cardiomyocytes in septic mice myocardial injury.

(A) Protein band plots and relative protein expression statistics of LC3I/II, Beclin and ATG5; (B) Transmission electron microscopy (TEM) observations. Data are presented as mean ± SEM. N = 10; **P < 0.01.

PTX3 inhibited the apoptotic capacity of cardiomyocytes in mice myocardial injury induced by sepsis

Western blot results showed that the protein expression levels of Bax, Caspase-3, and Caspase-9 in the cardiomyocytes of Groups B and C were higher than those of Group A, and the protein expression levels of Bcl-2 were lower than those of Group A (t-test showed P < 0.05). The expression levels of Bax, Caspase-3, and Caspase-9 proteins in the cardiomyocytes of Group C were lower than those in Group B, and the expression levels of Bcl-2 protein were higher than those in Group B (t-test showed P < 0.05). This indicated that PTX3 decreased the apoptotic capacity of cardiomyocytes in mice myocardial injury induced by sepsis (Fig. 5). The above results show that PTX3 can attenuate myocardial injury in sepsis due to the down-regulation of apoptosis and autophagy induced by the PI3K/AKT/mTOR pathway (Fig. 6).

Figure 5 PTX3 inhibited apoptotic capacity of cardiomyocyte with mice myocardial injury induced by sepsis.

(A) Protein band plots of Bax, Bcl-2, Caspase-3 and Caspase-9; (B) Relative protein expression statistics of Bax, Bcl-2, Caspase-3 and Caspase-9. Data are presented as mean ± SEM. N = 10; **P < 0.01.

Figure 6 PTX3 inhibits apoptosis caused by excessive autophagy by inhibiting the PI3K/AKT/mTOR signaling pathway, thereby alleviating the progression of sepsis.

Discussion

Manifesting as a systemic inflammatory response syndrome occasioned by infection, sepsis precipitates multi-organ dysfunction (Salomão et al., 2019). Myocardial injury is a frequent and grave complication of sepsis that considerably heightens patient morbidity and mortality (Frencken et al., 2018). The etiology of sepsis-induced myocardial injury, although elusive, appears intricately linked with the autophagic and apoptotic processes it incites (Hollenberg & Singer, 2021). This study has corroborated that PTX3 mitigates sepsis-provoked myocardial injury in mice through lentiviral-mediated overexpression and counteracts myocardial injury in LPS-induced septic mice. The underlying mechanism suppresses the expression of proteins associated with the PI3K/AKT/mTOR axis, which thereby restrains autophagy and apoptosis within cardiomyocytes.

Cardiac injury in septic conditions is predominately denoted by compromised systolic and diastolic functions of the ventricles (Elsawy et al., 2022). Research indicates PTX3 markedly elevates indices such as LVEF, LVFS, and LVAW thickness in d, as well as improves the E/A ratio and reduces the LVID, indicating that PTX3 has a certain improvement effect on systolic and diastolic dysfunction caused by septic myocardial injury.

Sepsis can initiate a cascade of deleterious cardiac alterations, including myocardial apoptosis (Shang et al., 2019). This pathological evolution is characterized not only by apoptotic and necrotic demise of varied cardiomyocytes, including those of mitochondrial origin (Zhang et al., 2022) triggered by sepsis, but is also marked by an augmented incursion of inflammatory cells into the myocardial tissue (Xiao et al., 2021). In this study, more mitochondrial destruction and autophagosomes occurred, inflammatory cell infiltration increased in the myocardium, and apoptosis was evident in the cardiomyocytes of mice in Group B, while these conditions improved in Group C mice. These results demonstrate that PTX3 can ameliorate cardiac pathological changes in septic myocardial injury.

Sepsis is a systemic inflammatory response syndrome due to infection that can lead to organ dysfunction and death. In the pathogenesis of sepsis, the PI3K/Akt/mTOR signaling pathway plays an important role. This pathway is a key intracellular signaling system that regulates cell survival, proliferation, metabolism, and immune and inflammatory responses. The modulation of the PI3K/AKT/mTOR pathway-related proteins has a profound influence on cellular apoptosis and autophagy (Zheng et al., 2021). Some studies have found that PI3K can promote the occurrence of autophagy (Xu et al., 2020). This study shows that the protein expression in Group C with high PTX3 expression is significantly lower than that in Group B, but higher than that in Group A via analyzing p-PI3K, p-AKT, and p-mTOR protein expression in mice cardiomyocytes. These results show that PTX3 inhibits the activation of the PI3K/AKT/mTOR pathway in septic myocardial injury.

Autophagy is a protective mechanism through which the cell can remove damaged organelles and proteins, helping the cell to restore and maintain balance. However, in the case of sepsis, autophagy may be disturbed, which leads to cell damage and death, which may eventually lead to multi-organ failure. When discussing the autophagy process in sepsis, LC3I/II, Beclin, and ATG5 are three commonly-mentioned autophagy-related proteins that play an important role in the autophagy pathway. Microtubule-associated protein 1A/1B light chain 3 (LC3) is a protein involved in autophagosome formation in autophagy. LC3 is available in two forms, LC3I and LC3II. LC3I is converted to LC3II through a phosphorylation process, and LC3II binds to the autophagosome membrane and is often used as a marker for autophagosome formation in the autophagic process. Beclin-1 is a protein encoded by the autophagy-related gene Beclin 1, which is another key factor in the initiation of autophagy. Beclin-1 interacts with Vps34, a specific phosphatidylinositol 3 kinase, to form a core complex that helps regulate the nucleation step of autophagosomes, the early formation process of autophagosomes. Autophagy-associated protein 5 (ATG5) is a key protein involved in the expansion and closure of autophagosomal membranes. By covalently binding to ATG12, ATG5 functions and binds to other ATG proteins to promote autophagosome maturation (Jiang et al., 2023). In comparison to the control group, the experimental group had lower expression levels of these markers, suggesting that PTX3 diminishes autophagic activity in septic cardiomyocytes.

Activation of autophagy is also associated with programmed cell death. Excessive autophagic activity sometimes leads to cell death or a phenomenon known as autophagic cell death, which is considered a different pathway of death from apoptosis and necrosis. In sepsis, the body’s immune system reacts to an infection, producing an inflammatory response. However, in the case of sepsis, this response is often excessive and can cause systemic inflammation, leading to tissue and organ damage. This systemic damage may lead to apoptosis, which is programmed cell death. Bax, Bcl-2, Caspase-3, and Caspase-9 play a crucial role in apoptosis, whose expression levels closely correlates to the apoptosis level of cells (Jia et al., 2021). Furthermore, we found that the protein expression levels of Bax, Caspase-3, and Caspase-9 were significantly lower, and the protein expression levels of Bcl-2 were significantly higher in Group B. These results indicate that PTX3 inhibits the apoptotic ability of cardiomyocytes in sepsis. Therefore, the above studies illustrate that in sepsis, PTX3 inhibits apoptosis caused by excessive autophagy by inhibiting the PI3K/AKT/mTOR signaling pathway, thereby alleviating the progression of sepsis. However, the animal models in this study can help us understand the mechanisms involved in myocardial injury in sepsis. However, whether the model can accurately represent the physiological processes in the human body still needs to be determined using human clinical trials in order to ensure its safety and efficacy. We will use the AAV virus with a specific promoter to analyze animal hearts in future experiments.

In conclusion, PTX3 inhibits apoptosis caused by excessive autophagy by inhibiting the PI3K/AKT/mTOR signaling pathway, thereby alleviating the progression of sepsis. This discovery lays the groundwork for potential clinical application in septic patients, forms the foundation for personalized medicine, and warrants exploration of PTX3’s synergism with other therapies to enhance treatment outcomes. Further, it aids in pinpointing pertinent biomarkers for assessing septic myocardial injury severity, predicting treatment responsiveness, and monitoring disease evolution and prognostication.

Supplemental Information

Supplemental Information 1 ARRIVE checklist.

Supplemental Information 2 Raw numerical data for animal & cell experiments (Figures 1-6).

Supplemental Information 3 Figure 3 Western Blot.

Supplemental Information 4 Figure 4 Western Blot.

Supplemental Information 5 Figure 5 Western Blot.

Additional Information and Declarations

Competing Interests

Author Contributions

Animal Ethics

Data Availability

The authors declare that they have no competing interests.

Na Cui conceived and designed the experiments, performed the experiments, prepared figures and/or tables, authored or reviewed drafts of the article, and approved the final draft.

Zhi Chen analyzed the data, prepared figures and/or tables, and approved the final draft.

Zhanbiao Yu analyzed the data, prepared figures and/or tables, and approved the final draft.

Xiaowei Lv analyzed the data, prepared figures and/or tables, and approved the final draft.

Zhenjie Hu conceived and designed the experiments, authored or reviewed drafts of the article, and approved the final draft.

The following information was supplied relating to ethical approvals (i.e., approving body and any reference numbers):

All procedures involving animals were carried out conforming to The Guide for Care and Use of Laboratory Animals and approved by the Experimental Animal Ethics Committee of Hebei University.

The following information was supplied regarding data availability:

The raw measurements are available in the Supplemental Files.

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
