# Peer review of "PTX3 mediates PI3K/AKT/mTOR signaling to downregulate apoptosis and autophagy to attenuate myocardial injury in sepsis"

_PeerJ, doi:10.7717/peerj.17263_

## Round 0.1 · original submission · Major Revisions

The paper investigates the effect and mechanism of PTX3 on myocardial injury in sepsis. The study design is well-structured and includes appropriate experimental models and methods. The findings of the study provide valuable information on the potential therapeutic effect of PTX3 on septic myocardial injury and the underlying mechanisms.

However, the paper lacks detailed information on the specific procedures and protocols used in the experiments, Can authors provide more information about experiments, such as the dosages of injected substances and the time points of measurement?

**Language Note:** The review process has identified that the English language must be improved. PeerJ can provide language editing services - please contact us at [email protected] for pricing (be sure to provide your manuscript number and title). Alternatively, you should make your own arrangements to improve the language quality and provide details in your response letter. – PeerJ Staff

Reviewer 1 ·

Basic reporting

The authors submitted a research article in which the reported that Down-regulation of apoptosis and autophagy induced by PI3K/AKT/mTOR pathway via PTX3 to attenuate myocardial injury in sepsis. The aim of the study is clear. The manuscript has a logical structure, which cover up all aspects of the hypothesis and the findings. The tables and figures are clear. The sections Discussion and Conclusions are clear and readable. I have no serious concerns of the paper, while I would like to make several minor comments:1.The grammar of the article needs to be slightly adjusted.

Experimental design

2.I noticed that the authors measured MV E and MV A in mice by echocardiography, and I was very curious about the effect of PTX3 gene on cardiac diastolic function in septic mice. Authors are advised to add the results for the E/A ratio, even if the result is negative.
3.I note that the authors edited the expression of the mouse cardiac PTX3 gene by lentivirus as a vector, which is theoretically feasible. Nevertheless, I recommend that the authors use an AAV virus with a specific promoter to edit animal hearts in future experiments.
4.The authors seem to have omitted the antibody information for p-PI3K in the methodological writing. It is suggested that the levels of t-AKT,t-PI3K and their corresponding phosphorylation in heart tissue should be detected at the same time. It is also suggested that the authors supplement the WB experiment of Fig5 with the detection of cardiac BCL2 levels.

Validity of the findings

-

Additional comments

-

Reviewer 2 ·

Basic reporting

no comment'

Experimental design

no comment'

Validity of the findings

no comment'

Additional comments

This paper titled as “Down-regulation of apoptosis and autophagy induced by PI3K/AKT/mTOR pathway via PTX3 to attenuate myocardial injury in sepsis” was well designed and the experiment process was complete. but the writting need more revision as bellows.
1: While some methods are mentioned, but lacks details. Provide more information about the study design, the number of participants, and the specific techniques used in the experiments. This will give readers a better understanding of the study's methodology.
2: Results: The results are briefly summarized but need more specific details. Mention actual numerical findings, statistical significance, and trends observed. This will help readers assess the significance of the results.
3: While you mention the establishment of a myocardial injury in sepsis model in vitro, provide more description or explanation of the model used. This can help readers understand the methodology.
4: At the end the introduction with a clear transition to the methods section. Briefly summarize what the study aims to achieve and how it will be carried out.
5: Animal Model: Describe the surgical procedures in establishing the sepsis model more clearly. This includes the techniques used, any specific tools or instruments, and how post-operative care and monitoring were carried out.
6: Statistical Significance: Where you report statistical significance (e.g., "P<0.05"), it's useful to indicate the statistical tests that were used. This helps readers understand the reliability of your findings.
7: Study limitations: Discuss the limitations of your study. Are there any factors that might have influenced the results or areas where further research is needed? Recognizing limitations demonstrates a thorough understanding of your study's scope.
8: Future directions: Suggest avenues for future research. What questions have arisen from your findings that could guide future investigations in the field of sepsis treatment ?
9: It is best to have uniform colours throughout the statistical charts.
10: English should be improved, especially improve grammar.
11: Modify the figure legends of the figures.You shall added what the means of the figure, not to say only the results of each figure.
12: Review references- several are very old.
13: Apoptosis experiment such as TUNEL or flow cytometry should be added in vitro experiments.
14: Redraw Figure 6, as current schematic is too rudimentary.

---

## Round 0.2 · accepted · Accept

Since the authors have fully addressed the comments, I have no more questions.

Reviewer 2 ·

Basic reporting

None.

Experimental design

None.

Validity of the findings

None.

Additional comments

This study aimed to investigate the effect and mechanism of Pentraxin 3 (PTX3) on myocardial injury in sepsis. The results showed PTX3significantly improved cardiac function and structure in sepsis-stricken mice, and PTX3alleviated cardiac damage caused by sepsis. PTX3reduced the relative protein expression of p-PI3K, p-AKT, mTOR, LC3I/II, Beclin, ATG5, Bax, Caspase-3, and Caspase-9 in septic mouse cardiomyocytes and increased the relative protein expression of Bcl-2. PTX3 can attenuate myocardial injury in sepsis due to the down-regulation of apoptosis and autophagy induced by the PI3K/AKT/mTOR pathway. After revision, the manuscript has greatly improved, including the hypothesis and the findings. The tables, figures, discussion and conclusions are also clear and readable. And therefore I suggest this paper could be accepted without further revision.